# The Utility of Pre-Treatment Inflammation Markers as Associative Factors to the Adverse Outcomes of Vulvar Cancer: A Study on Staging, Nodal Involvement, and Metastasis Models

**DOI:** 10.3390/jcm12010096

**Published:** 2022-12-22

**Authors:** Hariyono Winarto, Muhammad Habiburrahman, Tricia Dewi Anggraeni, Kartiwa Hadi Nuryanto, Renny Anggia Julianti, Gatot Purwoto, Andrijono Andrijono

**Affiliations:** 1Division of Gynecologic Oncology, Department of Obstetrics and Gynaecology, Faculty of Medicine, Universitas Indonesia, Dr. Cipto Mangunkusumo Hospital, Jakarta 10430, Indonesia; 2Department of Obstetrics and Gynaecology, Faculty of Medicine, Universitas Indonesia, Dr. Cipto Mangunkusumo Hospital, Jakarta 10430, Indonesia

**Keywords:** advanced stage, basophil-to-monocyte, body-mass-index, albumin, complete blood count, derived ratios, distant metastasis, inflammatory markers, lymph node metastasis, neutrophil-to-lymphocyte, vulvar cancer

## Abstract

Background: Given the role of inflammation in carcinogenesis, this study investigated the utility of pre-treatment inflammatory markers as associative indicators for advanced-stage disease, lymph node metastasis (LNM), and distant metastasis (DM) in vulvar cancer (VC). Methods: A cross-sectional study was conducted on 86 women with VC in a single centre in Jakarta, Indonesia. The laboratory data was based on C-reactive protein (CRP), procalcitonin, the erythrocyte sedimentation rate (ESR) and fourteen derived, recorded and calculated ratios: leukocyte-to-platelet (LPR), neutrophil-to-lymphocyte (NLR), derived neutrophil-to-lymphocyte (dNLR), neutrophil-to-monocyte (NMR), platelet-to-monocyte (PLR), lymphocyte-to-monocyte (LMR), basophil-to-monocyte (BLR), systemic immune-inflammation index (SII), body mass index, albumin, and NLR (BAN) score, haemoglobin-to-platelet (HPR), prognostic nutritional index (PNI), modified Glasgow Prognostic Score (mGPS), CRP-to-albumin, and CRP-to-procalcitonin. The optimal cut-off for each marker was determined using receiver operating characteristic (ROC) curve analysis, and their diagnostic indicator performances were assessed. The utility of these ratios as associative factors for three endpoints was further evaluated in multivariate regression models. Results: Investigated inflammatory markers exhibited specific performances for individual adverse outcomes, proving a fair to excellent ability in case finding and screening. After adjustment, the BAN score ≤ 334.89 (OR 9.20, *p* = 0.001) and ESR ≥ 104 (OR 4.18, *p* = 0.048) become two advanced-stage associative factors with AUC: 0.769. LNM was solely determined by higher NLR ≥ 2.83 (OR 4.15, *p* = 0.014) with AUC: 0.615. Meanwhile, BLR ≥ 0.035 (OR 5.67, *p* = 0.001) and ESR ≥ 84 (OR 6.01, *p* = 0.003) were contributing factors for DM, with AUC: 0.765. Conclusions: Inflammatory markers are crucial for identifying the deleterious outcomes of VC. Accordingly, yielded models require external validation.

## 1. Introduction

Vulvar malignancies are infrequent and primarily affect older women [1]. It comprises about 5% of all female genital tract malignancies and has become the fourth most common gynecologic cancer [2]. Vulvar carcinoma (VC) etiopathogenesis has been associated with local inflammatory processes such as human papillomavirus (HPV)-induced infections, lichen planus/sclerosus, psoriasis, allergies, and leucoplakia [1]. Proteomic analyses have also pointed to inflammation as a driver of progression, as evidenced by the presence of two inflammatory proteins (HMGA2 and PRTN3) in solid tissues and blood samples from patients with VC and premalignant vulvar lesions [3].

Accordingly, a subsequent disease-specific correlation between underlying systemic inflammatory milieu and clinical outcomes may be predicted. A recent study found systemic immune inflammation linked to improved accuracy in predicting survival in patients with aggressive VC [4]. Evaluating systemic inflammatory-based predictors in vulvar cancer may be of particular clinical relevance, as they may be related to staging and metastasis [5]. The advanced-stage disease bears a poor prognosis, where lymph node metastasis (LNM) and distant metastasis (DM) are the most critical prognostic parameters [2]. Occasionally, these parameters can be challenging to determine before surgery. Therefore, the quest for biological markers to assist in the prediction of tumour status, diagnosis, and prognosis of these patients is essential.

Numerous biomarkers have been examined on cancer patients to ascertain the patients’ clinicopathological status [6,7,8], one of which is pre-operative systemic inflammatory markers. This immune-inflammatory response to the neoplastic process is reflected by increased circulating pro-inflammatory cytokines, abundant leukocyte migration, and an elevated platelet count [9,10,11,12,13]. Several derived haematological markers are gaining interest as systemic inflammatory response (SIR) surrogate markers for various clinical situations. They can be calculated from a combination of complete blood count (CBC) results (i.e., haemoglobin, leukocytes and their differential counts, and platelets). These measurements were combined with body mass index (BMI), albumin, and well-known inflammatory markers (e.g., erythrocyte sedimentation rate, ESR; C-reactive protein, CRP; and procalcitonin, PCT), yielding an additional 14 surrogate markers [9,10,14,15].

The value of inflammatory markers has been assessed as prognostic factors in a variety of human cancers, including gynaecological malignancies like ovarian [16,17,18], cervical [19,20,21], endometrial [22], and uterine cancer [23]. However, only limited evidence has explored inflammatory indicators of VC; one study by Ertas et al. [7] solely assessed the role of neutrophil-to-lymphocyte ratio (NLR) and platelet-to-lymphocyte ratio (PLR) in determining lymph node involvement. However, that study did not research the use of inflammatory markers in predicting advanced clinical staging and DM. Furthermore, a thorough study on the clinical profile and baseline laboratory data of VC patients in Indonesia was not performed. Arising from these research gaps and preparedness to uncover a non-invasive and easy-to-use pre-operative test to estimate the patients’ clinical outcomes, we aimed to determine whether pre-treatment inflammatory markers can be used as associative markers for advanced-stage disease, LNM, and DM in patients with VC.

## 2. Materials and Methods

### 2.1. Study Design, Patients, and Eligibility Criteria

The consecutive databases of Indonesia’s leading national referral hospital, Dr Cipto Mangunkusumo Hospital (CMH), were reviewed retrospectively to identify patients with pathologically confirmed VC who underwent complete surgical staging between 1 January 2015, and 31 December 2020. The detailed steps of conducting this study, inclusion and exclusion criteria, and the sampling process executed up to data analysis have been illustrated in Figure 1. The research results were presented following the strengthening of the reporting of observational studies in epidemiology (STROBE) guidelines [24].

### 2.2. Study Variables: Patients Characteristics, Laboratory Results, and Markers Measurements

Patient data points were extracted from medical records, including age, BMI, LNM, DM, staging, and pre-operative laboratory results. Patients were categorised according to whether they had LNM status or not based on the outcomes of lymphadenopathy imaging (ultrasonography, computed tomography/CT, and/or magnetic resonance imaging/MRI) [2,25], which were then verified by pathology testing (cytology and/or histology) [26]. Imaging techniques were used to identify DM as distant cancer spread to other body parts (such as the lungs, liver, bones, and skin) and, if possible, pathology testing [27]. However, most of this investigation’s findings depended on imaging studies, such as CT, MRI, positron emission tomography and CT (PET-CT), PET-MRI, and chest X-ray, because not all instances could be validated by pathology [25,28]. Cancer staging was determined based on the International Federation of Gynaecology and Obstetrics (FIGO) 2021 revised classification system [29], except for vulvar melanoma, which were classified using the tumour, node, and metastasis (TNM) system [30].

All baseline blood parameters were obtained from the medical records retrospectively. The morning before treatment initiation (e.g., surgery), after an 8–10-h fast, peripheral blood samples (2 mL) of hospitalised patients were collected via phlebotomy of the cubital veins within two weeks before treatments [31,32] employing standard laboratory instruments. The measured laboratory data included haematology profiles, electrolytes, general chemistry analysis, enzyme analysis, protein analysis, haemostasis results, and clotting measurements. Table 1 describes several mathematical formulas that were used to calculate inflammatory markers.

### 2.3. Study Endpoints, Markers’ Cut-Off Determination, and Diagnostic Indicator Performances

Using SPSS, receiver operating characteristic (ROC) curve analysis was used to determine a cut-off based on the highest Youden index (maximum point of sensitivity and specificity) [43,44]. The study endpoints were models of staging, LNM status, and the presence of DM, and thus the cut-off of several ratios was tailored based on those three endpoints. The cut-offs indicate the presence of the endpoints with values higher/lower or equal to the cut-off. To determine the direction of testing, we referred to the previous literature. Leukocyte-to-platelet ratio (LPR), NLR, derived neutrophil-to-lymphocyte ratio (dNLR), neutrophil-to-monocyte ratio (NMR), PLR, basophil-to-monocyte ratio (BLR), systemic immune-inflammation index (SII), modified Glasgow prognostic score (mGPS), CRP/Alb ratio, and CRP/PCT ratio showed higher values than the cut-off, indicating expected cases with worse conditions. Meanwhile, a lower value than the cut-off indicated expected cases with worse conditions in the lymphocyte-to-monocyte ratio (LMR) [45], BMI, albumin, and NLR (BAN) score, haemoglobin-to-platelet ratio (HPR), and prognostic nutritional index (PNI) [45].

Two online medical calculators were used for diagnostic indicators performances. First, MedCalc v20.114 (Ostend, Belgium; 2022) [46] was used to confirm the sensitivity and specificity value resulting in ROC analysis previously and to determine the value of positive likelihood ratio (LR+), negative likelihood ratio (LR−), positive predictive value (PPV), negative predictive value (NPV), and accuracy. Meanwhile, the second calculator from Mitchell et al. [47,48] was used to determine the clinical utility index (CUI). Clinical utility is the degree to which a diagnostic test is helpful in clinical practice and comprises two indices: the CUI+ is a product of PPV and sensitivity, providing an indicator of the clinically relevant “rule in the cases” accuracy, and CUI− is a product of NPV and specificity, providing an indicator of the clinically relevant “rule out the cases” accuracy. Clinical utility depends on 3 factors: discrimination, occurrence and acceptability. The third is hard to quantify mathematically. Most diagnostic methods look at the first factor [49].

Several categories were applied in interpreting diagnostic performance indicators of the predetermined cut-off with the detailed information in Appendix A. The quality of the markers’ cut-off was appraised according to the area under the receiver operating characteristic curve (AUC) categorisation: “excellent” (0.9–1.0), “very good” (0.8–0.9), “good” (0.7–0.8), “sufficient” (0.6–0.7), “poor” (0.5–0.6), and test not useful or “worthless” (<0.5) [50]. The principle of “the lower the standard of error, the better the cut-off” was used according to ROC analyses. The sensitivity, specificity, PPV, NPV, and accuracy ratings were decided following a consensus: “excellent” (≥95%), “good” (80–95%), “moderate” (70–85%), and “poor” (<70%) [51]. Favourable results of LR+ are for values >10, and <0.1 for LR− [52]. The interpretation of CUI follows these rules of thumb: “excellent” (≥0.81), “good” (0.64–0.81), “fair” utility (0.49–0.64), “poor” utility (0.36–0.49), and “very poor” utility (<0.36) [47,48]. Generally, a CUI+ ≥ 0.49 indicates a test with abilities to identify certain endpoints accurately. A greater CUI+ value shows better power for ruling in (finding) a positive case. Meanwhile, a CUI− ≥ 0.49 indicates a test with screening abilities; a greater value reveals better power for ruling out (excluding) the negative case [47,48].

### 2.4. Statistical Analysis

The variables of patients’ characteristics, laboratory results, and inflammatory markers were reported as frequencies (percentages) for categorical variables and numerical descriptives (i.e., mean ± standard deviations, median, and interquartile range/IQR) for continuous variables. After the Kolmogorov-Smirnov or Shapiro-Wilk test was performed, normally distributed continuous data were analysed using the independent student *t*-test by considering Levene’s variance test results to determine the homogeneity of variances. The Mann–Whitney U-test was chosen if variables were not normally distributed. This study utilised the Spearman rank test because all marker data were not normally distributed when looking at the inter-surrogate marker associations and interactions. The correlation test’s interpretation of rho degree (*ρ*) followed the literature: 0, “no correlation” (0), “very weak” (0.01–0.2), “weak” (0.2–0.4), “moderate” (0.4–0.6), “strong” (0.6–0.8), “very strong” (0.8–1), and monotonic (1) [53].

The association of inflammatory markers and study endpoints (i.e., advanced-stage disease, LNM, and DM) were analysed via bivariate analysis using χ^2^ or Fisher’s exact tests with Mantel-Haenszel common odds ratio (OR). Variables that were identified as potential factors (*p* ≤ 0.25) for the groups of interest in the bivariate analysis (unadjusted analysis) were further analysed using a stepwise selection and backward multiple logistic regression (adjusted analysis) to produce an OR between the factors that contributed to the endpoints [54,55].

In order to evaluate the performance and externally validate the risk-factor model, the fit of the data to the model was calibrated using the Hosmer–Lemeshow test, and discrimination values were assessed using ROC and AUC [56]. The quality of the predictive model was classified based on the AUC value [50]. All statistical tests used SPSS v24.0 software (IBM, Chicago, IL, USA) and all analyses with a *p*-value less than 0.05 indicated statistical significance with a 95% confidence interval (CI). Visualisation of data employed SPSS v24.0 for Windows and Microsoft^®^ Excel^®^ from Microsoft Office 365 v.2207 32-bit (Redmond, WA, USA).

## 3. Results

### 3.1. Patients Characteristics

Eighty-six patients with VC were registered in the cancer registry of CMH. The median age of patients was 52.13 ±: 13.80. Half of the patients were in the age range between 41 and 60 years (50.0%), followed by these age groups: 61–80 years (31.4%), 21–40 years (15.1%), 0–20 years (2.3%), and 81–100 years (1.2%). The average BMI of the patients was 22.88 ± 4.65 kg/m^2^. According to the Asian classification for BMI, 39.5% of subjects had normal BMI (18.5–22.9 kg/m^2^), followed by obese (≥25 kg/m^2^) with 24.4% of patients, overweight (23–24.9 kg/m^2^) with 19.8% of patients, and with the minority of patients being underweight (<18.5 kg/m^2^) with 16.3% in proportion. Assessing the patients’ clinical characteristics, most patients had advanced-stage disease (81; 79.1%) and experienced DM (54; 62.8%). Meanwhile, the proportion of LNM occurrence was almost equal, with 45 (52.3%) patients having positive LNM.

### 3.2. Comparison of Baseline Laboratory Examination Results

The differences in laboratory baselines according to staging, LNM, and DM status are highlighted in Table 2 and Appendix A. It was revealed that advanced-stage VC had lower mean haemoglobin (*p* < 0.01), hematocrit (*p* < 0.05), and erythrocyte (*p* < 0.01) than its counterparts, along with a lower value of albumin (*p* < 0.01), as well as a lower percentage of basophils (*p* < 0.05) and lymphocytes (*p* < 0.05), but a higher percentage of neutrophils (*p* < 0.05). Unlike the former comparison, no evidence was found for the differences between present and absent LNM in laboratory results. Meanwhile, in comparing the DM subgroups, the difference was found solely in aspartate aminotransferase (AST), where higher concentrations of AST belonged to patients with DM (*p* < 0.05).

### 3.3. Comparison of Baseline Inflammatory Markers

Table 3 exhibits the findings of the comparison of inflammatory-associated cancer markers between staging, LNM and DM diagnosis. The analysis revealed significantly higher median values of NLR (*p* < 0.05), dNLR (*p* < 0.05), mGPS (*p* < 0.01), CRP (*p* < 0.01), procalcitonin (*p* < 0.05), and CRP/Alb ratio (*p* < 0.01) in advanced-stage VC, whereas significantly lower median values of LMR (*p* < 0.01), BAN score (*p* < 0.05), and PNI (*p* < 0.01) were observed in these patients. However, none of the seventeen marker differences was statistically significant in the two groups of LNM status. On the other hand, median ESR values were dramatically higher in patients with DM (*p* < 0.05).

**Table 2 jcm-12-00096-t002:** Baseline characteristics of laboratory testing results in the form of haematological, chemistry, and haemostasis/clotting analysis.

Laboratory Profiles	Overall Included Cases	Mean ± Standard Deviation or Median (Interquartile Range: 25–75% Quartile)								
		Clinical Staging			Lymph Node Metastasis (LNM)			Distant Metastasis (DM)		
		Early Stage/ I–II	Advanced Stage/III–IV	*p*-Value	LNM (−)	LNM (+)	*p*-Value	DM (−)	DM (+)	*p*-Value
Haemoglobin (g/dL)	10.6 ± 1.8	11.6 ± 2.1	10.5 ± 1.8	**0.005 ^a^**	10.8 ± 2.1	10.3 ± 1.5	0.226 ^b^	10.8 ± 1.8	10.1 ± 1.8	0.107 ^a^
Hematocrit (%)	31.4 ± 5.8	34.1 ± 7.7	30.7 ± 5.0	**0.029 ^a^**	31.9 ± 6.9	31.0 ± 4.6	0.522 ^b^	32.3 (28.3–35.8)	31.4 (28.1–34.1)	0.284 ^c^
Erythrocyte (×10^6^/μL)	3.8 ± 0.7	4.2 ± 0.8	3.7 ± 0.7	**0.009 ^a^**	3.8 ± 0.8	3.7 ± 0.6	0.512 ^b^	3.7 (3.5–4.3)	3.8 (3.3–4.1)	0.426 ^c^
MCV (fL)	83.8 ± 6.2	83.8 ± 7.9	83.8 ± 5.8	0.262 ^a^	83.9 ± 7.1	83.8 ± 5.5	0.952 ^a^	84.1 ± 5.7	83.4 ± 7.1	0.606 ^a^
MCH (pg)	28.4 (32.2–34.6)	28.4 (27.1–30.0)	28.4 (26.5–30.0)	0.920 ^c^	28.8 (26.9–30.0)	28.4 (26.4–29.7)	0.387 ^c^	28.6 (26.9–30.0)	28.1 (25.9–30.0)	0.447 ^c^
MCHC (g/dL)	33.5 ± 1.7	33.5 ± 1.4	33.6 ± 1.8	0.169 ^a^	33.7 ± 1.6	33.4 ± 1.8	0.545 ^a^	33.7 ± 1.5	33.3 ± 2.1	0.293 ^a^
Platelets (×10^9^/L)	346.9 ± 144.9	377.5 ± 144.5	338.7 ± 144.9	0.559 ^a^	347.1 ± 161. 6	346.6 ± 129.7	0.989 ^a^	349.8 ± 141.3	341.9 ± 152.9	0.808 ^a^
Leucocytes (count/μL)	10,085 (6850–15,492)	8865 (7737–12,807)	11,105 (6155–16,350)	0.832 ^c^	9840 (6840–16,540)	10,960 (6705–15,240)	0.952 ^c^	9365 (7040–15,402.5)	12,045 (5965–16,350)	0.865 ^c^
Basophils (%)	0.4 (0.2–0.5)	0.4 (0.3–0.6)	0.3 (0.2–0.5)	**0.047 ^c^**	0.4 (0.2–0.5)	0.3 (0.2–0.5)	0.686 ^c^	0.3 (0.2–0.5)	0.4 (0.2–0.5)	0.902 ^c^
Eosinophils (%)	1.5 (0.4–2.8)	1.2 (0.9–3.7)	1.5 (0.4–2.8)	0.625 ^c^	1.2 (0.4–2.2)	1.9 (0.4–2.9)	0.222 ^c^	1.4 (0.4–2.9)	1.6 (0.5–2.8)	0.844 ^c^
Neutrophils (%)	77.9 (69.0–84.1)	66.3 (60.5–81.2)	78.7 (71.5–84.7)	**0.047 ^c^**	77.9 (62.2–84.5)	77.9 (71.8–83.0)	0.749 ^c^	76.6 (66.4–83.4)	79.9 (70.0–84.7)	0.360 ^c^
Lymphocytes (%)	12.8 (6.8–20.2)	25.4 (11.6–28.4)	11.3 (6.2–16.6)	**0.011 ^c^**	12.9 (7.2–25.8)	12.8 (6.2–17.0)	0.397 ^c^	14.1 (7.3–25.2)	10.7 (5.5–15.2)	0.144 ^c^
Monocytes (%)	6.7 ± 2.4	6.1 (5.4–7.0)	6.8 (5.4–8.5)	0.364 ^c^	6.5 (5.4–8.3)	6.9 (5.3–8.5)	0.822 ^c^	6.5 ± 2.5	7.1 ± 2.1	0.270 ^a^
Ureum (mg/dL)	24.2 (18.6–35.5)	21.3 (18.2–28.8)	25.6 (18.9–41.0)	0.141 ^a^	24.0 (18.8–29.6)	25.2 (16.9–44.0)	0.331 ^a^	23.5 (17.4–38.7)	26.1 (19.3–32.7)	0.964 ^a^
Creatinine (mg/dL)	0.8 (0.6–1.0)	0.8 (0.7–0.9)	0.8 (0.6–1.1)	0.574 ^a^	0.7 (0.6–0.9)	0.8 (0.6–1.2)	0.139 ^a^	0.8 (0.7–1.0)	0.7 (0.6–0.9)	0.399 ^a^
eGFR (mL/min/1.73 m^2^)	84.0 (58.4–103.0)	83.8 (71.6–106.8)	85.9 (57.2–102.8)	0.411 ^c^	92.0 (71.7–107.8)	77.6 (46.5–102.0)	0.059 ^c^	77.1 ± 32.9	85.3 ± 26.1	0.233 ^a^
AST (µ/L)	19.0 (15.0–25.5)	21.0 (14.7–22.0)	19.0 (15.0–35.7)	0.648 ^c^	21.0 (16.0–30.5)	17.0 (14.0–23.5)	0.279 ^c^	18.5 (14.0–22.2)	22.5 (15.2–60.7)	**0.049 ^c^**
ALT (µ/L)	16.0 (11.0–23.0)	19.5 (10.7–22.7)	15.5 (11.0–23.0)	0.531 ^c^	18.0 (13.5–24.5)	13.00 (9.5–22.5)	0.094 ^c^	15.5 (10.0–22.0)	17.0 (11.0–33.7)	0.269 ^c^
Albumin (g/dL)	3.3 (2.7–4.0)	4.0 (3.4–4.2)	3.2 (2.6–3.8)	**0.003 ^c^**	3.4 ± 0.7	3.2 ± 0.8	0.152 ^a^	3.3 ± 0.8	3.1 ± 0.8	0.277 ^a^
Patient PT (seconds)	10.5 (10.2–11.5)	10.3 (10.1–10.8)	10.7 (10.3–11.8)	0.066 ^c^	10.4 (10.0–11.3)	10.7 (10.3–11.9)	0.194 ^c^	10.5 (10.1–11.4)	10.7 (10.2–11.8)	0.497 ^c^
Patient aPTT (seconds)	34.2 (29.3–38.4)	33.5 (29.4–35.9)	34.7 (29.0–39.1)	0.545 ^c^	35.5 (30.2–39.0)	33.2 (28.8–38.0)	0.268 ^c^	33.9 (29.4–38.3)	35.8 (29.0–38.7)	0.678 ^c^

^a^ *t*-test with equal variances assumed; ^b^ *t*-test with equal variances not assumed; ^c^ Mann-Whitney U test. ALT, alanine aminotransferase; AST, aspartate aminotransferase; aPTT, activated partial thrombin time; eGFR, estimated glomerular filtration rate; MCH, mean corpuscular haemoglobin; MCHC, mean corpuscular haemoglobin concentration; MCV, mean corpuscular volume; PT, prothrombin time.

### 3.4. Intercorrelations between Inflammatory Markers

An intercorrelation analysis between the 17 marker measures was performed to assess further the influence of the inflammatory markers on VC (Figure 2). Together, these results provide important insights into intercorrelations between inflammatory markers. SII and procalcitonin appeared to be prominent inflammatory indicators since they correlated the most with other markers (14 significant correlations from 17 tests). On the contrary, HPR is the marker with the fewest correlations to other markers (2 significant correlations from 17 tests). CRP and the CRP/Alb ratio appeared to have the strongest positive correlations (*ρ* = 0.986, *p* < 0.001). When only considering inflammatory surrogate markers derived from calculations, the strongest positive correlation was between NLR and dNLR (*ρ* = 0.950, *p* < 0.001). On the other hand, LPR and BLR had the weakest positive correlation among the individual inflammatory markers (*ρ* = 0.232, *p* < 0.05). The results of the correlational analysis also indicated that the most negative correlation between the two markers was between the NLR and BAN score (*ρ* = −0.956, *p* < 0.001), whereas the interaction between the NMR and PNI (*ρ* = −0.384, *p* < 0.001) had the least negative correlations. Among 3 well-known inflammatory markers, moderate associations were discovered between ESR and CRP (*ρ* = 0.461, *p* < 0.01) and CRP and procalcitonin (*ρ* = 0.456, *p* < 0.05); meanwhile, procalcitonin did not affect ESR (*ρ* = 0.244, *p* > 0.05).

### 3.5. Determination of Inflammatory Markers Cut-Off and Their Diagnostic Performance

Optimal cut-off values for pre-treatment inflammatory markers (Table 4 and Appendix A) were tailored to their association with three disease outcomes using ROC analysis attached in Appendix A such as Appendix A (for clinical staging), Appendix A (for LNM), and Appendix A (for DM). Each cut-off has been measured along with individual diagnostic performance indicators. In general, most markers had LR+ <10 and LR− >0.1, which indicates poor performance as predictors. Hence, they should only be regarded as covariates [57].

According to AUC values, three of the seventeen studied markers had excellent category performances and were associated with advanced clinical stages. They were CRP, procalcitonin, and the CRP/Alb ratio. One marker, mGPS, was very good; two were good (LMR and PNI score), and seven were deemed sufficient (NLR, dNLR, PLR, BLR, SII, BAN score, and ESR). The AUC of the markers HPR, LPR, NMR, and the CRP/PCT ratio was regarded as unreliable. In terms of clinical practice utility, the higher the CUI+ value, the better the power to rule cases in (confirming incidents). With this regard, our findings revealed that CRP, procalcitonin, and the CRP/Alb ratio were three markers with excellent utility for finding and detecting the cases. NLR, dNLR, SII, BAN score, PNI score, and mGPS were ‘good’; meanwhile, LPR, PLR, and LMR were regarded as ‘fair’ markers. On the other hand, a higher CUI− indicates a test that can rule out cases; the higher the value, the better the test’s ability to do so. Given the highest CUI− value, only CRP was good at screening out the cases; others exhibited poor and very poor clinical utility. 

In model-associated factors of LNM, it was deduced that most markers had low performance according to their AUC values. Most were not helpful markers to discriminate the presence of LNM. However, procalcitonin was a marker with a sufficient AUC value. A poor capability of inflammatory markers in association with the detection of LNM was also proven by CUI analyses. The utility markers with a ‘fair’ rating to determine whether a case has LNR were NLR, dNLR, PNI score, CRP, and CRP/Alb ratio. Procalcitonin and the CRP/PCT ratio had a fair utility in excluding the cases of LNM based on the CUI− indicator. Meanwhile, the clinical utility of other markers was poor to extremely poor.

Regarding DM model metrics, no good AUC values were found for all markers. LMR, BLR, BAN score, ESR, PNI score, CRP, and CRP/Alb were in the highest category, which was merely ‘average’. Meanwhile, it was discovered that the AUC values of LPR, NLR, dNLR, PLR, SII, HPR, mGPS, procalcitonin, and CRP/PCT ratio were low, indicating poor discriminating capabilities. Another ratio, NMR, revealed the lowest values among all the markers connected to DM incidence. BAN score, PNI score, CRP, and CRP/Alb ratio showed sufficient AUC values, although their *p*-value was insignificant (*p* > 0.05). This means that only LMR, BLR, and ESR had a significant ability to distinguish DM occurrence (*p* < 0.05). Clinically, the identified markers typically have poor case detection and DM screening discriminatory capacities. As evidenced by the most significant value of CUI+, only procalcitonin had fair case-finding capabilities, and CUI− values showed that only BLR, HPR, CRP, and the CRP/Alb ratio had average case-excluding abilities.

### 3.6. Association of Inflammatory Markers with Clinical Staging Endpoints

Cut-offs were employed to stratify eligible patients into two groups (low and high). In bivariate analysis, it was noted that groups with high NLR, dNLR, PLR, BLR, SII, ESR, CRP, and procalcitonin, and groups with low LMR, BAN score, and PNI groups had significantly higher association with advanced-stage cancer. However, not all of them are included in the multivariable logistic regression models due to zero values in the tabulation. Only two out of nine examined markers—the BAN score and the ESR—were significant factors related to advanced-stage disease (Table 5 and Appendix A). A lower BAN score exhibited up to 9.20 times higher odds of being diagnosed with advanced-stage disease. Meanwhile, ESR possessed an OR of 4.18 (*p* = 0.048) for advanced-stage determination. This model demonstrated ‘good’ classification abilities in the ROC testing (Figure 3A), with an AUC value of 0.769 (*p* < 0.0001).

### 3.7. Association of Inflammatory Markers with Lymph Node Metastasis Endpoints

The results showed six significant markers connected to the occurrence of LNM in comparing low- and high-value categories of markers (Table 6 and Appendix A). The presence of LNM was associated with high NLR, dNLR, BLR, and procalcitonin, as well as lower scores of BAN and PNI in the unadjusted analysis. After adjustment in multivariate logistic regression analysis, the result yielded one independent associated factor, NLR, with a 315% increase in the odds of detection of LNM with a given exposure. The logistic regression ROC analysis (Figure 3B) demonstrated that this model was an “average” discriminator, with an unremarkable AUC value (0.615, *p* = 0.066), indicating that NLR was not excellent in distinguishing negative and positive LNM cases.

### 3.8. Association of Inflammatory Markers with Distant Metastasis Endpoints

Cases were stratified into two classes according to the presence of DM, and bivariate analyses were conducted according to the markers’ cut-offs (Table 7 and Appendix A). Higher levels of NLR, BLR, SII, ESR, CRP, procalcitonin, and CRP/Alb, as well as lower levels of LMR, BAN score, and PNI score were found as related factors to DM. After performing a multivariate logistic regression analysis, it was established that BLR and ESR were two critical indicators for detecting DM. BLR and ESR had OR values of 5.67 (*p* = 0.001) and 6.01 (*p* = 0.003), respectively. The model was then examined using ROC analysis, and an AUC value of 0.765 (*p* < 0.0001) indicated that it accurately represented DM discrimination (Figure 3C).

**Figure 3 jcm-12-00096-f003:**
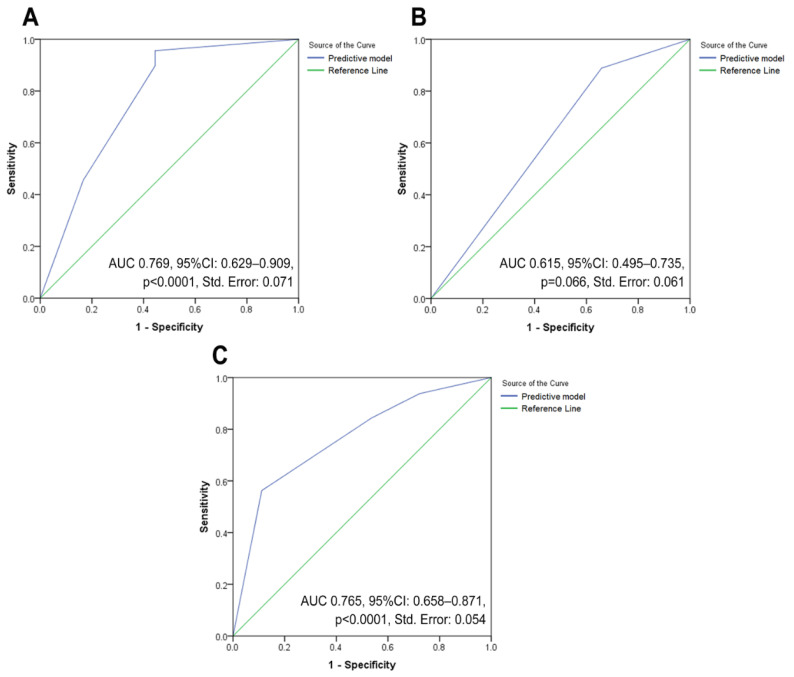
Three ROC analysis curves employed to assess the ability of significant inflammatory markers in modelling three clinical endpoints of vulvar cancer: (**A**) performance evaluation for a model that includes factors associated with advanced-stage vulvar cancer; (**B**) performance assessment for a model that takes into account factors related to the incidence of lymph node metastases in vulvar cancer patients; and (**C**) performance estimation for a model that include factors related to the incidence of distant metastases in vulvar cancer patients. **Abbreviation:** AUC: area under the curve; CI: confidence interval; ROC, receiver operating characteristic curve.

## 4. Discussion

The wide intracellular array of signalling pathways is often deregulated during inflammation, resulting in malignant transformation through genomic instability induction, DNA damage, and promotion of cell proliferation [58]. Inflammatory markers were importantly proven predictive of unfavourable prognosis of gynaecological malignancies [59,60,61], including VC [7]. This study has tested several inflammatory markers associated with the advanced-stage of VC. Lower pre-operative BAN score became one of two significant associative factors of advanced stage disease (adjusted odd ratio, aOR, 9.20, *p* = 0.001). The BAN score has been proposed as a novel indicator for assessing the nutritional status and ongoing systemic inflammation response in various advanced malignancies, including oesophagal [37] and gastric cancer [62,63]. It is hypothesised that patients with low BAN scores are more prone to be malnourished, leading to worse oncological outcomes [64]. Low scores indicate a decrease in BMI and albumin with an increase in NLR, which might be responsible for aggressive tumour biology, cancer progression, and poor prognosis [65]. Our examination revealed a greater percentage of hypoalbuminemia in advanced-stage VC, corresponding to a discovery made in a study on epithelial ovarian cancers study (Table 2) [66]. Tumour cells can inhibit the liver’s ability to synthesise albumin by releasing inflammatory cytokines and/or promoting invasion and metastasis [67,68]. This discovery is consistent with our results, which show that DM patients had greater alanine aminotransferase levels (ALT), possibly due to liver metastasis (Table 2).

Moreover, being underweight, which was observed in 16.3% of our patients, might reduce serum albumin levels and cause lymphocytopenia [69,70,71]. Lymphocytopenia is also associated with ageing, cancer severity, and poorer prognosis [72]. This fact is confirmed by our finding in Table 2, where half of our patients were elderly, which increases the risk of malnutrition and worsens T lymphocyte function [73,74]. Unsurprisingly, the median percentage of lymphocytes was significantly lower in patients with advanced stage than in their counterparts (11.35% vs. 25.45%, *p* = 0.011). This evidence indicates that malnutrition and lymphocytopenia may serve as indicators of the chronically impaired immune system and may collectively promote advanced development in VC.

We also discovered that higher ESR (≥104 mm/h) became the second associative factor linked to an advanced-stage VC (aOR 4.18, *p* = 0.048). The DM model further demonstrated the associated significance of ESR ≥84 mm/h in the clinical outcomes of VC (aOR 6.01, *p* = 0.003). Similar to a prior study, this work found that a high ESR level was associated with metastatic disease but not LNM [75]. The cut-offs of ESR used in this study were comparable to those used in other studies (80–100) [52,53]. ESR is a marker of chronic inflammatory conditions [76] and a significant predictor of cancer-specific survival of solid tumours [77,78,79]. It was demonstrated that a first-time hospital diagnosis of increased ESR is a strong indicator of discovering undetected cancer in the first 12 months of following up. Elevated ESR was also associated with an 8.5% chance of developing cancer within the first year after contact and significantly worse survival than matched cancer comparisons with a mortality rate ratio of 1.2 [80]. ESR is a cheap and practical laboratory test that is useful for diagnosing and monitoring various chronic conditions (e.g., cancer). In comparison to expenses for testing CRP, which were around $6.73–20.20 (in Indonesia), and $28.00–219.00 (in the US), ESR costs were cheaper: $1.35–2.02 (in Indonesia) and $4–22 (in the US). ESR, as demonstrated here, thus has the advantage of being relatively inexpensive in addition to its clinical relevance.

With ESR, higher BLR emerged as one of the significant factors linked to DM (aOR 5.67, *p* = 0.001). However, the multivariate analysis was unsatisfactory in the clinical stages and LNM models. Cut-offs for BLR in the staging and DM models were ≥0.035, somewhat higher in the LNM model (≥0.045), but these cut-offs corresponded with the cut-off value for cervical cancer in a prior Indonesian study (≥0.034) [20]. There are a limited number of studies about BLR in the cancer literature; previous studies reported basophil ratios to have prognostic roles in various types of cancer, such as cervical and pancreatic cancers [19,20,81]. Since basophils are closely related to chronic inflammation, there might also be a significant role of BLR in VC [82]. As another component of BLR, infiltrating lymphocytes can reduce tumours’ growth, eradicate them, and trigger an antitumour cellular immune response [83,84]. Therefore, when the high ratio is brought on by a high basophil count but a low lymphocyte count, this could reflect an inadequate immunological reaction to the tumour and, consequently, a weakened defence against cancer [85,86].

The LNM associative factors model was the most challenging to develop since many inflammatory markers yielded insignificant associations with LNM occurrences. Nevertheless, multivariate analyses proved that NLR markers were the sole factor associatable with LNM (aOR 4.15, *p* = 0.014). We found that LNM was more common in the high NLR group than in their counterparts (88.9% vs. 11.1%, *p* = 0.010), corroborated by a previous study [7], which revealed the significant association between NLR > 2.81 and LNM in VC with the corresponding values 60.7% vs. 5.6%, *p* < 0.001, and aRR 10.90, *p* < 0.001). NLR is an inflammation- and immunity-related marker reflecting neutrophilia and lymphopenia to evaluate cancer patients’ clinical outcomes and is also used in cervical cancer [59] and ovarian cancer [87]. The recruitment of neutrophils, which promotes cancer cell invasion, migration, and angiogenesis, was a plausible explanation for their role in LNM [88]. However, in the staging and DM models, their ability failed to be proven when encountering multivariate analyses. In this study, NLR shared the same cut-off value of ≥2.83 for the LNM and staging models; however, in DM, the value was ≥5.67. This value was in line with those utilised in the research on breast cancer (>2.88) [89] and ovarian cancer (>6) [90].

Unlike NLR, multivariate analysis in this study did not reveal any proof of a connection between dNLR and staging, LNM, or DM. NLR and dNLR look identical and have similar effects on cancer-specific mortality [91,92]. Our investigation supported this hypothesis, which found a robust positive correlation between dNLR and NLR (*ρ* = 0.950, *p* < 0.001). Bivariate analysis revealed that the high dNLR case groups had a larger proportion of advanced-stage cases (85.3% vs. 14.7%, OR 7.25, *p* = 0.001). Additionally, it was found that the median dNLR in the advanced stage case group was higher than in the early stage case group (3.71 vs. 1.97, *p* = 0.047). Likewise, the group with the higher level of dNLR had more cases of LNM than their counterparts (86.7% vs. 13.3%, OR 3.37, *p* = 0.022). However, the level of dNLR and the presence of DM patients indicated no unique relationship (*p* = 0.082). There is currently no information on the prognostic importance of this marker in VC, despite studies showing that elevated dNLR was associated with poor prognoses in other malignancies [93].

Other markers revealed a non-significant role after adjustment in the multivariate analyses, one of which was LPR. This inflammatory marker was investigated in many diseases associated with non-cancer systemic inflammation [33,94] but is scarcely discussed in cancer [95]. Therefore, its clinical impact was still conflicted, and there is no evidence supporting the use of this marker in gynaecological cancer cases [33,94,95,96]. Nevertheless, the aggressive nature of tumours with marked leukocytosis (an element of LPR) occurred in our baseline CBC results (Table 2). In this study, we found an insignificantly higher level of LPR among VC cases with advanced stage, LNM and DM, indicating the low performance of this parameter. The second marker with dubious aptitude as an associative factor of VC outcomes is PLR. Although high PLR was associated with advanced stage disease (OR 4.48, *p* = 0.006) in bivariate analysis, it failed to be significant in multivariate analysis. It was also shown that the rate of LNM was slightly more remarkable in the high PLR group than in the low PLR group (62.2% vs. 37.8%, *p* = 0.089), partially agreeing with earlier VC studies that discovered a significant difference (54.8% vs. 6.1%, *p* < 0.001) [7]. In contrast to our findings, a study from Turkiye showed that PLR was a significant predictor of LNM with a OR 10.4, *p* = 0.008 [7]. Extreme thrombocytosis may promote carcinogenesis and cancer metastasis. On the other hand, by secreting thrombopoietic cytokines and tumour-derived platelet factor 4 (PF4), tumour cells induce thrombopoiesis [97,98,99]. Meanwhile, lymphocyte has the role of secreting interleukin (IL)-2, which inhibits tumour cell proliferation by activating and stimulating the proliferation of cytotoxic lymphocytes [100]. Thus, when the lymphocyte count turns low, this mechanism increases cancer progression leading to poor outcomes [7,101,102].

Up to now, only a few studies have analysed the prognostic value of LMR in non-haematological malignancies [103], one of which was in cervical cancer [20]. In a prognostic melanoma model, Schmidt et al. showed that an elevated monocyte count might act as an independent prognostic factor for poor survival [104]. In our study, the multivariate analysis failed to produce statistically significant results when testing this marker in three models. However, in bivariate analysis, low LMR was associated with higher odds of having advanced-stage VC (OR of 7.32, *p* = 0.001). It was also shown that the median LMR value was lower in the advanced stages (1.77 vs. 2.84, *p* = 0.007). In bivariate analysis, LMR was also related to DM (OR 4.02, *p* = 0.006) but was not linked with LNM (OR 2.34, *p* = 0.052). Further, neither DM nor LNM was associated with LMR in multivariate analyses.

Since many inflammatory processes play a significant role in the development of VC, there may be a disease-specific relationship between SII levels and VC. SII, a novel inflammatory measure based on the number of peripheral lymphocytes, neutrophils, and platelets, was shown to improve prognosis prediction of gynaecological cancers in previous studies [105,106,107]. SII is noteworthy because it correlated with almost all mentioned inflammatory markers. A higher percentage of advanced-stage cases was observed in the high SII case groups (76.5% vs. 23.5%, OR 4.06, *p* = 0.009), corroborated by a prior study that discovered a positive correlation between SII and increased tumour stage in endometrial [22], and colorectal cancers [108]. Similarly, the higher SII group had more cases of DM than their counterparts (84.4% vs. 15.6%, OR 3.44, *p* = 0.023). Meanwhile, the presence of LNM and the degree of SII did not significantly differ (*p* = 0.093), in contrast with a meta-analysis of patients with gynaecological cancers (but not including VC), which revealed that greater SII was linked to an increased risk of LNM [109]. Nevertheless, SII failed to be a significant factor in the multivariate analyses in the three outcome models for VC in this study.

On the other hand, although being markers similarly used to predict inflammation and nutritional status [110,111,112], PNI, mGPS, and CRP/Alb ratio also failed to be decisive factors associated with the VC outcomes in the three adjustment models. Low PNI scores were associated with advanced stage (OR 6.65, *p* < 0.0001), LNM (OR 2.96, *p* = 0.028), and DM (OR 2.79, *p* = 0.034) only in bivariate analysis. Similarly, a higher score of mGPS had only a significant association with more advanced staging (*p* < 0.010) in bivariate analysis. On the other hand, we tested the CRP/Alb ratio in predicting the outcome of VC because this novel marker has gained significant attention recently for its ability to detect malnutrition and systemic inflammation in ovarian [112] and cervical cancers [113]. The higher median CRP/Alb ratio observed in advanced-stage patients compared to the early-stage patients (10.28 vs. 0.98, *p* = 0.005) supports an earlier study associating CRP/Alb with a more aggressive disease phenotype in ovarian cancer [112]. This higher aggressivity was also seen in our study, where a higher CRP/Alb ratio was substantially linked to the development of DM (OR 6.12, *p* = 0.036). However, the role of CRP/Alb was unsatisfactory in identifying the LNM and advanced stage of the disease, even in bivariate analysis.

Other valuable but insignificant results in the three final adjustment models of VC were CRP, procalcitonin, and the CRP/PCT ratio. Today, there is no proof that CRP directly encourages the growth of VC. Recent findings in breast [114] and ovarian cancers [115] proposed CRP as an independent factor for predicting their outcomes. In Figure 2, CRP was found to be moderately linked with ESR and procalcitonin, demonstrating its significance in inflammation, but it might not be as strong as ESR in carcinogenesis. The median CRP levels also increased as VC progressed (*p* = 0.005). An investigation in ovarian cancer cases [116] revealed a similar correlation between high CRP levels and tumour stage (*p* < 0.0001), suggesting inflammation happened in both cancers [117]. In bivariate analysis, we detected a significant correlation (OR 13.12, *p* = 0.015) between high CRP levels and the occurrence of DM (but not with LNM), which is consistent with findings in a previous study among patients with metastatic CRC [118].

Similar to CRP, few studies highlighted the role of procalcitonin in advanced gynaecological malignancies [119], one of which was a study on ovarian cancer [120]. We discovered that the median value of procalcitonin was more remarkable in advanced-stage VC than in the early stage (0.67 vs. 0.05, *p* = 0.044). The patient group with higher procalcitonin showed a significantly larger proportion of patients with advanced stage than their counterparts (86.2% vs. 13.8%, *p* = 0.032). Additionally, we found that the DM group had substantially more patients with higher procalcitonin levels than those with lower levels (93.3% vs. 6.7%, *p* = 0.037). This outcome is in line with the research on solid tumours, which indicates that procalcitonin levels in patients with metastases are considerably greater than in those without metastases [121,122]. Interestingly, LNM occurrences were more prevalent in the low procalcitonin group (35.3% vs. 64.7%, *p* = 0.021), indicating that procalcitonin had a more critical role as a predictor of the advancement of neoplastic disease in the presence of DM, rather than LNM. Of note, CRP and procalcitonin could also become a combination or marker, which is the CRP/PCT ratio. This marker is the most rarely studied in cancer and has only been studied in solid tumours [42]. The CRP/PCT ratio was suitable for discriminating between infection-related and cancer-related fever [42]. Given their rarity, it is not surprising that the weak CRP/PCT potential in three endpoints was demonstrated in our bivariate and ROC analysis.

Together with the CRP/PCT ratio, the weakest performance markers for predicting VC outcomes in this study were HPR and NMR. These markers had the poorest CUI in identifying VC with advanced stage, LNM, and DM and were the least correlated with other inflammatory markers. Moreover, HPR and NMR had an insignificant association with advanced stage, LNM, and DM occurrence in bivariate and multivariate analyses. Therefore, they were not suggested as helpful markers for predicting clinical outcomes of VC. The low performance of NMR might be due to monocyte and neutrophil composition not being dramatically altered in our VC patients; thus, the expectation of significantly elevated NMR did not happen. On the contrary, a pancreatic cancer study suggests that elevated NMR is independently associated with poor prognosis [36]. The underlying idea of using HPR was based on the finding of concomitant anaemia and thrombocytosis in our VC patients. In our advanced-stage cases, the mean score for haemoglobin, haematocrit, and erythrocyte count was significantly lower, showing more prominent anaemia in the late stage of the disease. Cancer-associated anaemia might increase hypoxia-inducible factor-1 (HIF-1) and enhance the production of vascular endothelial growth factor (VEGF) [123]. As a result, these processes trigger the up-regulation of angiogenesis-related genes, which suppress apoptosis and promote the spread of cancer cells [123,124]. However, our investigation identified no significant effects in any of the three models, in contrast to a CRC study that found a link between low HPR and high tumour stage, LNM, and cancer invasion [125,126].

In light of all findings, this work proposes that the clinical impact of the BAN score and ESR would be to stratify patients with VC according to their risks based on their systemic inflammatory status, enabling caregivers to decide the complexity of VC cases [127]. Given the poor prognosis associated with LNM, discussions based on pre-treatment NLR results may help the individual patient consider whether LNM might have been or would be present. Furthermore, as predicted by increased pre-treatment BLR and ESR, in the context of DM, they may be utilised to determine if it is worth the risk to apply a more conservative therapeutic approach or palliative care [128].

### Strengths and Limitations

This work is the first study to compare various blood inflammatory markers with detrimental clinical features (stage, nodal involvement, and DM) in Indonesian women with VC, a disease that is rarely reported. This study also investigated the diagnostic performance indicators of 17 inflammatory markers, resulting in a thorough preliminary reference value for cut-offs and their abilities to determine VC outcomes. Additionally, the information was gathered over an extended period in the largest referral hospital in the nation, which may partially reflect the general population.

The present study is, however, subject to several limitations. The rarity of VC renders the design of any well-controlled prospective study difficult. Thus, this study employs a retrospective cross-sectional design, typically lacking randomisation and possibly flawed data acquisition. Because it relied on the (retrospectively gathered) medical records, we could not control the process and timing of blood collection and analysis. Additionally, the number of patients within our registry data is limited. Pre-operative CRP and procalcitonin data were also limited because these tests were not performed routinely at our centre. Consequently, those parameters and their derivations are ineligible to be included in multivariate analyses. Despite all the drawbacks, this study could be a preliminary to a more extensive, valid, and randomised study. Additionally, although inexpensive and easy, applying use of haematological markers in clinical practice could be challenging because of a lack of standardisation and evidence regarding clinically validated thresholds in VC [129]. Therefore, inflammatory indicators should be used more often, given their substantial value in cancer diagnostics and prognostics.

## 5. Conclusions

In conclusion, pre-treatment inflammation markers indicate an essential association with adverse clinical outcomes of VC. The examined inflammatory markers have unique associative values at particular cut-offs tailored for specific endpoints. They are generally of fair to excellent utility as case finders and screening markers and could play a substantial role in managing patients with VC. Our study highlights the BAN score and ESR as reliable markers associated with cases of advanced-stage disease. The utility of NLR was also superior and became the sole factor influencing LNM. Furthermore, BLR and ESR provide essential information on the risk stratification of cases with DM. Our findings align with the justification for conducting prospective multi-centric studies to validate the clinical use of these markers and incorporate this research into the management and prognosis of VC patients.

## Figures and Tables

**Figure 1 jcm-12-00096-f001:**
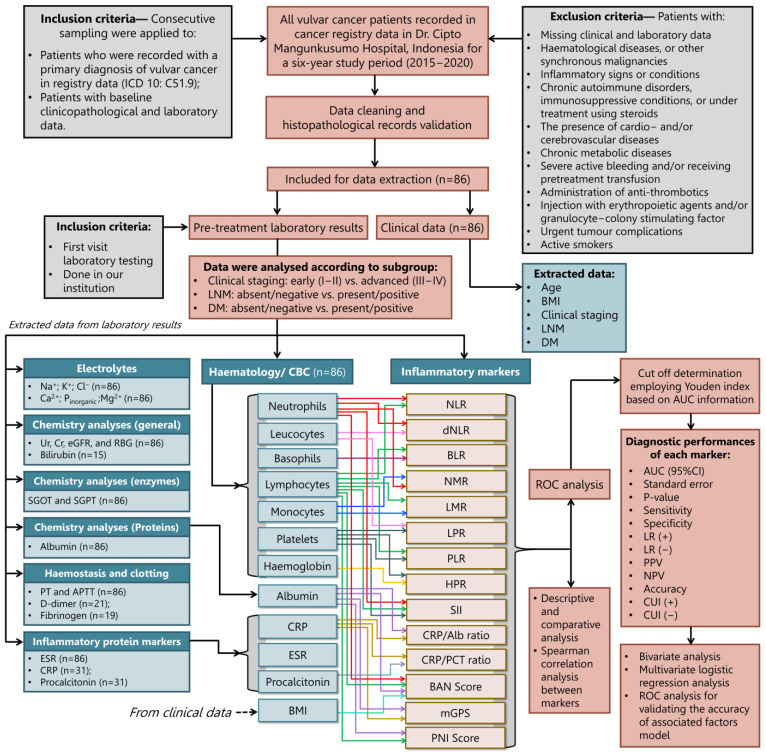
The study workflow describes patient enrolment and inclusion, followed by the analytical data pipeline of subgrouping according to the patient’s clinical staging, lymph node metastasis, and distant metastasis.

**Figure 2 jcm-12-00096-f002:**
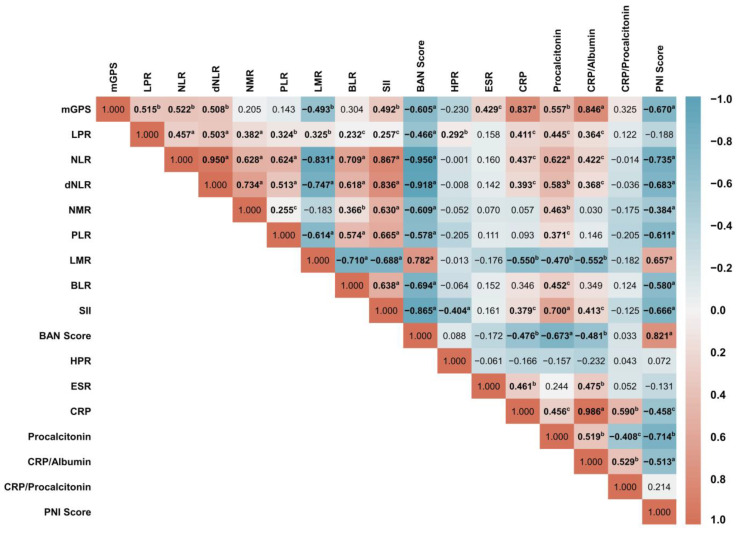
The correlation matrix between inflammatory surrogate marker metrics was depicted as a heatmap, measured using Spearman rank correlation analysis. The heat map represents the colour-coded correlation factors between all markers. The colour value of the cells is proportional to the strength of the associations, ranging from red (positive correlations) to blue (negative correlations). The strength of the correlation is indicated in the colour scale (at the right of the panel), where 1 shows a strong positive correlation, −1 shows a strong negative correlation, and 0 shows no correlation. Pair-wise-Spearman correlation coefficients (*ρ*) are shown in every cell. A *p*-value correlation between two inflammatory markers was described as ^a^ *p* < 0.001; ^b^ *p* < 0.01; and ^c^ *p* < 0.05, and others without superscripts were denoted as non-significant (*p* > 0.05).

**Table 1 jcm-12-00096-t001:** The formulas of inflammatory surrogate markers.

No	Surrogate Markers	Abbreviation	Formula	Ref.
1.	Leukocyte-to-platelet ratio	LPR	$\frac{\mathrm{leukocytes}\;(\mathrm{count}/\mu L)}{\mathrm{platelets}\;(\mathrm{count}/\mu L)}$	[33]
2.	Neutrophil-to-lymphocyte ratio	NLR	$\frac{\mathrm{neutrophils}\;(\mathrm{count}/\mu L)}{\mathrm{lymphocytes}\;(\mathrm{count}/\mu L)}$	[34]
3.	Derived neutrophil-to-lymphocyte ratio	dNLR	$\frac{\mathrm{neutrophils}\;(\mathrm{count}/\mu L)}{\left[\mathrm{leucocytes}\;(\mathrm{count}/\mu L)-\mathrm{neutrophils}\;(\mathrm{count}/\mu L)\right]}$	[34,35]
4.	Neutrophil-to-monocyte ratio	NMR	$\frac{\mathrm{neutrophils}\;(\mathrm{count}/\mu L)}{\mathrm{monocytes}\;(\mathrm{count}/\mu L)}$	[36]
5.	Platelet-to-lymphocyte ratio	PLR	$\frac{\mathrm{platelets}\;(\mathrm{count}/\mu L)}{\mathrm{lymphocytes}\;(\mathrm{count}/\mu L)}$	[34]
6.	Lymphocyte-to-monocyte ratio	LMR	$\frac{\mathrm{lymphocytes}\;(\mathrm{count}/\mu L)}{\mathrm{monocytes}\;(\mathrm{count}/\mu L)}$	[34]
7.	Basophil-to-monocyte ratio	BLR	$\frac{\mathrm{basophils}\;(\mathrm{count}/\mu L)}{\mathrm{lymphocytes}\;(\mathrm{count}/\mu L)}$	[19]
8.	Systemic immune-inflammation index	SII	$\frac{\mathrm{neutrophils}\;(\mathrm{count}/\mu L)}{\mathrm{lymphocytes}\;(\mathrm{count}/\mu L)}{\;\times \;\mathrm{platelets}\;(\mathrm{in}\;10}^{9}\;\mathrm{cells}/L)\;$	[7]
9.	Body-mass-index, albumin and NLR score	BAN score	$\mathrm{BMI}\;\left({\mathrm{kg}/m}^{2}\right)\;\times \;\frac{\mathrm{albumin}\;(g/\mathrm{dL})}{\frac{\mathrm{neutrophils}\;(\mathrm{count}/\mu L)}{\mathrm{lymphocytes}\;(\mathrm{count}/\mu L)}}\;$	[37,38]
10.	Haemoglobin-to-platelet ratio	HPR	$\frac{\mathrm{haemoglobin}\;(g/L)}{\mathrm{platelets}\;(\mathrm{count}/\mu L)}{\;(\mathrm{the}\;\mathrm{ratio}\;\mathrm{is}\;\mathrm{in}\;10}^{9}\;\mathrm{cells}/L)$	[39]
11.	Prognostic nutritional index	PNI	[10 × albumin (g/dL)] + [0.005 × lymphocytes (count/μL)]	[40]
12.	Modified Glasgow prognostic score	mGPS	Scoring ranges from 0 to 2: CRP (>10 mg/L) and hypoalbuminemia (<3.5 g/dL) = 2 CRP (>10 mg/L), but normal value for albumin = 1 Normal CRP level and albumin level = 0	[41]
13.	CRP-to-albumin ratio	CRP/Alb ratio	$\frac{\mathrm{CRP}\;\mathrm{level}\;\left(\mathrm{mg}/L\right)}{\mathrm{albumin}\;\mathrm{level}\;\left(g/\mathrm{dL}\right)}{(\mathrm{the}\;\mathrm{ratio}\;\mathrm{is}\;\mathrm{in}\;10}^{-4})$	[38]
14.	CRP-to-procalcitonin ratio	CRP/PCT ratio	$\frac{\mathrm{CRP}\;\mathrm{level}\;\left(\mathrm{mg}/L\right)}{\mathrm{procalcitonin}\;\mathrm{concentration}\;\left(\mathrm{ng}/\mathrm{mL}\right)}{(\mathrm{the}\;\mathrm{ratio}\;\mathrm{is}\;\mathrm{in}\;10}^{3})$	[42]

**Table 3 jcm-12-00096-t003:** Baseline characteristics of inflammatory marker calculation results.

Inflammatory Markers	Overall Included Cases	Mean ± Standard Deviation or Median (Interquartile Range: 25–75% Quartile)								
		Clinical Staging			Lymph Node Metastasis (LNM)			Distant Metastasis (DM)		
		Early Stage/ I–II	Advanced Stage/ III–IV	*p*-Value	LNM (−)	LNM (+)	*p*-Value	DM (−)	DM (+)	*p*-Value
LPR *	29.80 (20.30–50.25)	25.95 (20.17–47.65)	30.80 (21.00–52.27)	0.510 ^a^	30.80 (20.30–51.80)	29.00 (20.10–47.10)	0.789 ^a^	26.65 (21.72–48.70)	37.15 (18.85–52.27)	0.789 ^a^
NLR *	6.27 (3.35–11.69)	2.60 (2.17–7.10)	6.74 (4.51–13.20)	**0.014 ^a^**	6.19 (2.38–11.83)	6.37 (4.23–12.31)	0.421 ^a^	5.45 (2.63–11.22)	7.44 (4.91–15.53)	0.421 ^a^
dNLR *	3.52 (2.23–5.31)	1.97 (1.53–4.34)	3.71 (2.51–5.53)	**0.047 ^a^**	3.52 (1.65–5.47)	3.52 (2.55–4.88)	0.749 ^a^	3.28 (1.98–5.02)	3.97 (2.33–5.53)	0.749 ^a^
NMR *	10.99 (8.77–14.90)	10.99 (9.05–14.35)	11.06 (8.76–15.15)	0.920 ^a^	11.13 (9.24–13.95)	10.65 (8.71–15.53)	0.986 ^a^	10.99 (9.01–15.74)	11.06 (8.08–13.54)	0.986 ^a^
PLR *	251.73 (164.87–449.27)	175.1 (156.3–259.4)	262.0 (175.7–495.3)	0.075 ^a^	244.8 (162.7–353.6)	264.5 (172.3–545.6)	0.202 ^a^	246.8 (164.8–450.5)	259.2 (163.6–458.6)	0.202 ^a^
LMR *	1.89 (1.17–2.92)	2.84 (1.97–4.29)	1.77 (1.10–2.67)	**0.007 ^a^**	2.19 (1.16–3.08)	1.76 (1.16–2.94)	0.497 ^a^	2.25 (1.21–3.75)	1.70 (0.97–2.21)	0.497 ^a^
BLR *	0.03 (0.02–0.05)	0.02 (0.17–0.32)	0.03 (0.02–0.05)	0.153 ^a^	0.03 (0.02–0.04)	0.03 (0.02–0.06)	0.526 ^a^	0.03 (0.02–0.04)	0.04 (0.02–0.06)	0.526 ^a^
SII *	2178.0 (1030.3–4006.0)	1211.9 (836.0–2883.5)	2263.8 (1371.9–4140.5)	0.087 ^a^	1571.3 (970.3–3346.0)	2256.2 (1362.7–4270.5)	0.310 ^a^	1915.1 (932.2–4119.1)	2332.2 (1440.0–3778.4)	0.310 ^a^
BAN score *	111.52 (51.20–274.82)	396.12 (71.72–541.26)	105.34 (46.77–185.00)	**0.015 ^a^**	115.81 (49.30–396.12)	104.67 (53.73–189.34)	0.371 ^a^	136.32 (63.30–348.67)	100.83 (37.76–165.74)	0.371 ^a^
HPR *	0.30 (0.23–0.45)	0.33 (0.22–0.44)	0.29 (0.23–0.48)	0.795 ^a^	0.34 (0.23–0.48)	0.28 (0.23–0.43)	0.613 ^a^	0.32 (0.24–0.45)	0.28 (0.21–0.50)	0.613 ^a^
ESR *	95.00 (69.25–121.25)	79.00 (50.75–100.75)	105.00 (71.00–122.00)	0.076 ^a^	87.00 (58.00–118.50)	105.00 (77.00–122.00)	0.343 ^a^	85.52 ± 36.47	102.12 ± 27.49	**0.019 ^b^**
PNI score *	41.03 ± 12.80	49.50 (39.75–54.25)	39.00 (32.00–45.00)	**0.001 ^a^**	41.00 (34.50–50.00)	39.00 (32.00–45.50)	0.165 ^a^	42.50 (34.00–50.00)	38.50 (32.00–43.75)	0.131 ^a^
mGPS **	2.00 (0.00–2.00)	0.00 (0.00–0.00)	2.00 (0.00–2.00)	**0.009 ^a^**	1.50 (0.00–2.00)	2.00 (0.00–2.00)	0.913 ^a^	1.50 (0.00–2.00)	2.00 (0.00–2.00)	0.913 ^a^
CRP **	22.60 (6.11–165.90)	3.95 (1.42–4.67)	41.10 (8.10–176.70)	**0.005 ^a^**	39.60 (3.97–195.87)	18.10 (6.95–120.75)	0.606 ^a^	16.46 (4.22–80.35)	41.10 (6.30–190.70)	0.606 ^a^
Procalcitonin **	0.45 (0.15–1.96)	0.05 ^c^	0.67 (0.18–2.29)	**0.044 ^a^**	0.32 (0.12–1.10)	0.67 (0.17–3.85)	0.293 ^a^	0.31 (0.05–3.00)	0.67 (0.29–1.10)	0.293 ^a^
CRP/Alb ratio **	9.63 (1.78–61.90)	0.98 (0.37–1.14)	10.28 (2.25–68.04)	**0.005 ^a^**	10.07 (0.99–77.92)	9.58 (1.85–34.64)	0.691 ^a^	8.34 (1.04–21.52)	10.28 (2.25–87.48)	0.691 ^a^
CRP/PCT ratio ***	157.44 (31.45–273.05)	124.72 ^c^	157.44 (22.89–343.61)	>0.999 ^a^	192.18 (54.44–544.29)	64.57 (14.62–233.36)	0.284 ^a^	77.28 (51.11–555.25)	171.94 (18.13–273.05)	0.284 ^a^

^a^ Mann-Whitney U test; ^b^ *t*-test with equal variances not assumed; ^c^ only two cases available; * *n* = 86; ** *n* = 31; *** *n* = 24. **Abbreviations:** BAN, body mass index, albumin and neutrophil-lymphocyte ratio; BLR, basophil-to-monocyte ratio; CRP, C-reactive protein; CRP/Alb ratio, C-reactive protein-to-albumin ratio; CRP/PCT ratio, C-reactive protein-to-procalcitonin ratio; dNLR, derived neutrophil-to-lymphocyte ratio; ESR, erythrocyte sedimentation rate; HPR, haemoglobin-to-platelet ratio; LMR, lymphocyte-to-monocyte ratio; LPR, leukocyte-to-platelet ratio; mGPS, modified Glasgow Prognostic Score; NLR, neutrophil-to-lymphocyte ratio; NMR, neutrophil-to-monocyte ratio; PCT, procalcitonin; PLR, platelet-to-lymphocyte ratio; PNI, prognostic nutritional index; SII, systemic immune-inflammation index.

**Table 4 jcm-12-00096-t004:** Cut-offs and diagnostic indicators performance of inflammatory markers.

Diagnostic Performance Indicators	Inflammatory Markers																
	LPR	NLR	dNLR	NMR	PLR	LMR	BLR	SII	BAN	HPR	ESR	PNI	mGPS	CRP	Procalcitonin	CRP/Alb	CRP/PCT
**Clinical staging**																	
Cut-off	22.70	2.83	2.075	15.765	202.14	2.205	0.035	1348.115	334.89	0.325	104	47.50	0.5	5.485	0.11	1.295	228.52
AUC	0.55	0.69	0.65	0.51	0.64	0.71	0.61	0.63	0.69	0.48	0.64	0.75	0.87	0.94	0.93	0.94	0.50
*p*-value	0.510	**0.014**	**0.047**	0.920	0.075	**0.007**	0.158	0.087	**0.015**	0.795	0.076	**0.001**	**0.018**	**0.005**	**0.044**	**0.005**	>0.999
Sensitivity	72.06%	86.76%	85.29%	23.53%	69.12%	67.65%	48.53%	76.47%	89.71%	52.94%	51.47%	80.88%	74.07%	92.59%	86.21%	92.60%	31.80%
Specificity	44.44%	55.56%	55.56%	88.89%	66.67%	77.78%	77.78%	55.56%	55.56%	55.56%	83.33%	61.11%	100%	100%	100%	100%	100%
CUI+	0.598	0.764	0.750	0.209	0.613	0.622	0.433	0.663	0.793	0.433	0.474	0.718	0.741	0.926	0.862	0.871	0.318
CUI−	0.132	0.292	0.278	0.209	0.242	0.302	0.222	0.214	0.327	0.132	0.260	0.280	0.364	0.667	0.333	n/a	0.118
**Lymph node metastasis**																	
Cut-off	24.65	2.83	2.075	14.315	248.985	1.89	0.045	1413.135	238.45	0.325	87.5	47.50	1.50	5.485	2.72	1.295	880.665
AUC	0.48	0.55	0.52	0.50	0.58	0.54	0.54	0.56	0.56	0.53	0.56	0.59	0.51	0.44	0.61	0.46	0.37
*p*-value	0.789	0.421	0.749	0.986	0.202	0.497	0.531	0.310	0.371	0.613	0.344	0.165	0.921	0.606	0.293	0.691	0.284
Sensitivity	66.67%	88.89%	86.67%	33.33%	62.22%	60.00%	37.78%	75.56%	82.22%	60.00%	66.67%	82.22%	52.94%	94.12%	35.29%	94.12%	15.38%
Specificity	39.02%	34.10%	34.10%	78.05%	56.1-%	60.98%	82.93%	41.46%	36.59%	58.54%	51.22%	39.02%	50.00%	35.71%	100%	35.71%	100%
CUI+	0.364	0.531	0.512	0.208	0.379	0.377	0.268	0.443	0.483	0.368	0.400	0.491	0.298	0.602	0.353	0.602	0.154
CUI−	0.201	0.252	0.239	0.403	0.323	0.355	0.455	0.252	0.239	0.334	0.299	0.260	0.233	0.298	0.560	0.298	0.500
**Distant Metastasis**																	
Cut-off	34.15	5.67	3.455	9.535	223.965	2.34	0.035	1348.115	183.84	0.235	84	43.50	0.50	164.4	0.16	53.245	122.525
AUC	0.52	0.59	0.56	0.45	0.54	0.63	0.65	0.55	0.60	0.53	0.63	0.60	0.55	0.68	0.56	0.68	0.51
*p*-value	0.806	0.169	0.360	0.464	0.514	0.039	**0.020**	0.432	0.133	0.636	**0.047**	0.131	0.621	0.082	0.540	0.086	0.954
Sensitivity	56.25%	68.75%	65.62%	65.62%	68.75%	81.25%	65.62%	84.38%	84.38%	37.50%	84.38%	75.00%	73.33%	46.67%	93.33%	46.67%	66.67%
Specificity	62.96%	53.70%	53.70%	38.89%	48.15%	48.15%	70.37%	38.89%	40.74%	77.78%	46.30%	48.15%	43.75%	93.75%	43.75%	87.50%	58.33%
CUI+	0.266	0.322	0.300	0.255	0.303	0.391	0.372	0.380	0.386	0.188	0.407	0.346	0.403	0.408	0.568	0.363	0.410
CUI−	0.446	0.399	0.389	0.255	0.348	0.391	0.546	0.314	0.332	0.527	0.386	0.368	0.278	0.611	0.383	0.557	0.371

**Abbreviations:** AUC: area under the curve; BAN, body mass index, albumin and neutrophil-lymphocyte ratio; BLR, basophil-to-monocyte ratio; CI: confidence interval; CRP, C-reactive protein; CRP/Alb ratio, C-reactive protein-to-albumin ratio; CRP/PCT ratio, C-reactive protein-to-procalcitonin ratio; CUI: clinical utility index; dNLR, derived neutrophil-to-lymphocyte ratio; ESR, erythrocyte sedimentation rate; HPR, haemoglobin-to-platelet ratio; LMR, lymphocyte-to-monocyte ratio; LPR, leukocyte-to-platelet ratio; mGPS, modified Glasgow Prognostic Score; NLR, neutrophil-to-lymphocyte ratio; NMR, neutrophil-to-monocyte ratio; PCT, procalcitonin; PLR, platelet-to-lymphocyte ratio; PNI, prognostic nutritional index; SII, systemic immune-inflammation index.

**Table 5 jcm-12-00096-t005:** Performance of inflammatory markers using their tailored cut-offs associated with clinical staging.

Inflammatory Markers	Clinical Staging		Total	Bivariate Analysis		Multivariate Analysis	
	Advanced	Early		Unadjusted OR (95%CI)	*p*-Value	Adjusted OR (95%CI)	*p*-Value
High LPR (≥22.70)	49 (72.1%)	10 (55.6%)	59 (68.6%)	2.06 (0.71–6.01)	0.180 ^b,c^	1.04 (0.20–5.40)	0.961 ^d^
High NLR (≥2.83)	59 (86.8%)	8 (44.4%)	67 (77.9%)	8.19 (2.56–26.26)	**<0.0001 ^a,c^**	2.71 (0.19–38.14)	0.460 ^d^
High dNLR (≥2.075)	58 (85.3%)	8 (44.4%)	66 (76.7%)	7.25 (2.30–22.82)	**0.001 ^a,c^**	Not Defined (0)	>0.999 ^d^
High NMR (≥15.765)	16 (23.5%)	2 (11.1%)	18 (20.9%)	2.46 (0.51–11.87)	0.339 ^a^	**Not analysed**	
High PLR (≥202.14)	47 (69.1%)	6 (33.3%)	53 (61.6%)	4.48 (1.48–13.54)	**0.006 ^b,c^**	1.93 (0.43–8.65)	0.389 ^d^
Low LMR (≤2.205)	46 (67.6%)	4 (22.2%)	50 (58.1%)	7.32 (2.16–24.83)	**0.001 ^b,c^**	3.80 (0.77–18.70)	0.100 ^d^
High BLR (≥0.035)	33 (48.5%)	4 (22.2%)	37 (43.0%)	3.33 (1.00–11.05)	**0.045 ^b,c^**	0.73 (0.12–4.64)	0.741 ^d^
High SII (≥1348.115)	52 (76.5%)	8 (44.4%)	60 (69.8%)	4.06 (1.37–12.03)	**0.009 ^b,c^**	Not Defined (0)	>0.999 ^d^
Low BAN score (≤334.89)	61 (89.7%)	8 (44.4%)	69 (80.2%)	10.89 (3.23–36.71)	**<0.0001 ^a,c^**	9.20 (2.61–32.45)	**0.001 ^d^**
Low HPR (≤0.325)	36 (52.9%)	8 (44.4%)	44 (51.2%)	1.41 (0.49–4.00)	0.521 ^b^	**Not analysed**	
High ESR (≥104)	35 (51.5%)	3 (16.7%)	38 (44.2%)	5.30 (1.41–20.00)	**0.008 ^b,c^**	4.18 (1.01–17.32)	**0.048 ^d^**
Low PNI score (≤47.50)	55 (80.9%)	7 (38.9%)	62 (72.1%)	6.65 (2.16–20.46)	**<0.0001 ^b,c^**	1.43 (0.10–20.90)	0.794 ^d^
High mGPS (1–2)	20 (74.1%)	0	20 (64.5%)	n/a	**0.010 ^a^**	**Not analysed**	
High CRP (≥5.485)	25 (92.6%)	0	25 (80.6%)	n/a	**<0.0001 ^a^**	**Not analysed**	
High PCT (≥0.11)	25 (86.2%)	0	25 (80.6%)	n/a	**0.032 ^a^**	**Not analysed**	
High CRP/Alb (≥1.295)	27 (100%)	4 (100%)	31 (100%)	n/a	n/a	**Not analysed**	
High CRP/PCT (≥228.52)	7 (31.8%)	0	7 (29.2%)	n/a	>0.999 ^a^	**Not analysed**	

^a^ Fisher’s exact test; ^b^ χ^2^ test; OR was obtained from the Mantel-Haenszel common odds ratio estimate; ^c^ variables with *p*-value ≤ 0.25 were eligible to enter multivariate analysis after bivariate analysis, except variables with any n/a results for their OR; ^d^ multivariate analysis using the backward model; “n/a (not applicable)” denoted an incalculable OR due to the presence of invalid (null) data in the 2 × 2 table; percent values (%) were calculated as a percentage of the column total.

**Table 6 jcm-12-00096-t006:** Performance of inflammatory markers using their tailored cut-offs associated with lymph node metastasis.

Inflammatory Markers	LNM		Total	Bivariate Analysis		Multivariate Analysis	
	LNM (+)	LNM (−)		Unadjusted OR (95%CI)	*p*-Value	Adjusted OR (95%CI)	*p*-Value
High LPR (≥24.65)	30 (66.7%)	25 (61.0%)	55 (64.0%)	1.28 (0.53–3.09)	0.583 ^a^	**Not analysed**	
High NLR (≥2.83)	40 (88.9%)	27 (65.9%)	67 (77.9%)	4.15 (1.34–12.87)	**0.010 ^a,c^**	4.15 (1.34–12.86)	**0.014 ^d^**
High dNLR (≥2.075)	39 (86.7%)	27 (65.9%)	66 (76.7%)	3.37 (1.15–9.87)	**0.022 ^a,c^**	Not Defined (0)	>0.999 ^d^
High NMR (≥14.315)	15 (33.3%)	9 (22.0%)	24 (27.9%)	1.78 (0.68–4.67)	0.240 ^a,c^	1.25 (0.38–4.19)	0.712 ^d^
High PLR (≥248.985)	28 (62.2%)	18 (43.9%)	46 (53.5%)	2.10 (0.89–4.98)	0.089 ^a,c^	1.00 (0.29–3.50)	0.989 ^d^
Low LMR (≤1.89)	27 (60.0%)	16 (39.0%)	43 (50.0%)	2.34 (0.99–5.57)	0.052 ^a,c^	1.74 (0.50–6.08)	0.387 ^d^
High BLR (≥0.045)	17 (37.8%)	7 (17.1%)	24 (27.9%)	2.95 (1.07–8.11)	**0.033 ^a,c^**	2.05 (0.70–6.03)	0.192 ^d^
High SII (≥1413.135)	34 (75.6%)	24 (58.5%)	58 (67.4%)	2.19 (0.87–5.50)	0.093 ^a,c^	0.39 (0.09–1.77)	0.222 ^d^
Low BAN score (≤238.45)	37 (82.2%)	26 (63.4%)	63 (73.3%)	2.67 (1.00–7.21)	**0.049 ^a,c^**	0.31 (0.02–3.82)	0.359 ^d^
Low HPR (≤0.325)	27 (60.0%)	17 (41.5%)	44 (51.2%)	2.12 (0.89–5.01)	0.086 ^a,c^	2.00 (0.82–4.90)	0.127 ^d^
High ESR (≥87.5)	30 (66.7%)	20 (48.8%)	50 (58.1%)	2.10 (0.88–5.02)	0.093 ^a,c^	1.77 (0.69–4.55)	0.237 ^d^
Low PNI score (≤47.50)	37 (82.2%)	25 (61.0%)	62 (72.1%)	2.96 (1.10–7.96)	**0.028 ^a,c^**	1.26 (0.25–6.29)	0.780 ^d^
High mGPS (2)	9 (52.9%)	7 (50.0%)	16 (51.6%)	1.12 (0.27–4.63)	0.870 ^a^	**Not analysed**	
High CRP (≥5.485)	16 (94.1%)	9 (64.3%)	25 (80.6%)	8.89 (0.89–88.40)	0.067 ^b^	**Not analysed**	
High PCT (≥2.72)	6 (35.3%)	0	6 (19.4%)	n/a	**0.021 ^b^**	**Not analysed**	
High CRP/Alb (≥1.295)	16 (94.1%)	9 (64.3%)	25 (80.6%)	8.89 (0.89–88.40)	0.067 ^b^	**Not analysed**	
High CRP/PCT (≥880.665)	2 (15.4%)	0	2 (8.3%)	n/a	0.482 ^b^	**Not analysed**	

^a^ χ^2^ test; ^b^ Fisher’s exact test; OR was obtained from the Mantel-Haenszel common odds ratio estimate; ^c^ variables with *p*-value ≤ 0.25 were eligible to enter multivariate analysis after bivariate analysis, except variables with any n/a results for their OR. Only variables with the same sample size (*n* = 86) were included in this analysis; ^d^ multivariate analysis using the backward model; “n/a (not applicable)” denoted an incalculable OR due to the presence of invalid (null) data in the 2 × 2 table; percent values (%) were calculated as a percentage of the column total.

**Table 7 jcm-12-00096-t007:** Performance of inflammatory markers using their tailored cut-offs associated with distant metastasis.

Inflammatory Markers	Distant Metastasis (DM)		Total	Bivariate Analysis		Multivariate Analysis	
	DM (+)	DM (−)		Unadjusted OR (95%CI)	*p*-Value	Adjusted OR (95%CI)	*p*-Value
High LPR (≥34.15)	18 (56.3%)	20 (37.0%)	38 (44.2%)	2.19 (0.90–5.32)	0.083 ^a,c^	1.06 (0.30–3.81)	0.922 ^d^
High NLR (≥5.67)	22 (68.8%)	25 (46.3%)	47 (54.7%)	2.55 (1.02–6.40)	**0.043 ^a,c^**	0.71 (0.17–3.03)	0.648 ^d^
High dNLR (≥3.455)	21 (65.6%)	25 (46.3%)	46 (53.5%)	2.21 (0.90–5.47)	0.082 ^a,c^	0.84 (0.12–5.86)	0.859 ^d^
High NMR (≥9.535)	21 (65.6%)	33 (61.1%)	54 (62.8%)	1.21 (0.49–3.02)	0.676 ^a^	**Not analysed**	
High PLR (≥223.965)	22 (68.8%)	28 (51.9%)	50 (58.1%)	2.04 (0.81–5.12)	0.125 ^a,c^	0.81 (0.22–2.93)	0.747 ^d^
Low LMR (≤2.34)	26 (81.3%)	28 (51.9%)	54 (62.8%)	4.02 (1.42–11.34)	**0.006 ^a,c^**	1.94 (0.51–7.45)	0.332 ^d^
High BLR (≥0.035)	21 (65.6%)	16 (29.6%)	37 (43.0%)	4.53 (1.78–11.54)	**0.001 ^a,c^**	5.67 (2.02–15.87)	**0.001 ^d^**
High SII (≥1348.115)	27 (84.4%)	33 (61.1%)	60 (69.8%)	3.44 (1.14–10.32)	**0.023 ^a,c^**	0.89 (0.07–11.12)	0.930 ^d^
Low BAN score (≤183.84)	27 (84.4%)	32 (59.3%)	59 (68.6%)	3.73 (1.24–11.13)	**0.015 ^a,c^**	1.34 (0.18–9.90)	0.772 ^d^
Low HPR (≤0.235)	12 (37.5%)	12 (22.2%)	24 (27.9%)	2.10 (0.80–5.49)	0.127 ^a,c^	1.90 (0.62–5.83)	0.260 ^d^
High ESR (≥84)	27 (84.4%)	29 (53.7%)	56 (65.1%)	4.65 (1.56–13.90)	**0.004 ^a,c^**	6.01 (1.81–19.91)	**0.003 ^d^**
Low PNI score (≤43.50)	24 (75.0%)	28 (51.9%)	52 (60.5%)	2.79 (1.06–7.29)	**0.034 ^a,c^**	0.82 (0.16–4.11)	0.810 ^d^
High mGPS (1–2)	11 (64.7%)	9 (64.3%)	20 (64.5%)	2.14 (0.47–9.70)	0.320 ^a^	**Not analysed**	
High CRP (≥164.4)	7 (46.7%)	1 (6.3%)	8 (25.8%)	13.12 (1.36–126.30)	**0.015 ^b^**	**Not analysed**	
High PCT (≥0.16)	14 (93.3%)	9 (56.3%)	23 (74.2%)	10.89 (1.14–103.98)	**0.037 ^b^**	**Not analysed**	
High CRP/Alb (≥53.245)	7 (46.7%)	2 (12.5%)	9 (29.0%)	6.12 (1.01–36.89)	**0.036 ^a^**	**Not analysed**	
High CRP/PCT (≥122.525)	8 (66.7%)	5 (41.7%)	13 (54.2%)	2.80 (0.53–14.73)	0.219 ^a^	**Not analysed**	

^a^ χ^2^ test; ^b^ Fisher’s exact test; OR was obtained from the Mantel-Haenszel common odds ratio estimate; ^c^ variables with *p*-value ≤ 0.25 were eligible to enter multivariate analysis after bivariate analysis, except variables with any n/a results for their OR. Only variables with the same sample size (*n* = 86) were included in this analysis; ^d^ multivariate analysis using the backward model; “n/a (not applicable)” denoted an incalculable OR due to the presence of invalid (null) data in the 2 × 2 table; percent values (%) were calculated as a percentage of the column total.

## Data Availability

The relevant additional data is available from the corresponding author upon reasonable request, given approval provided by our University’s Institutional Review Board.

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
