# Peer review of "The Utility of Pre-Treatment Inflammation Markers as Associative Factors to the Adverse Outcomes of Vulvar Cancer: A Study on Staging, Nodal Involvement, and Metastasis Models"

_jcm, 2022, doi:10.3390/jcm12010096_

Round 1

Reviewer 1 Report

The data presented in the manuscript are relevant to expand knowledge about inflammatory factors in vulvar cancer. I suggest making adjustments to the formatting of the tables.

Author Response

Dear Reviewer 1,
The authors truly respect the time and effort you dedicated to providing constructive and insightful feedback on our manuscript. We appreciate the valuable input which allows us to revise our manuscript.
We outline our responses in answering the reviewer's comments and the detailed explanations in the attached document below. All the reviewer comments help us improve our manuscript and make it valuable to readers. 

Reviewer 2 Report

The manuscript is well-written, with explanatory tables and images. Exhaustive description of the study design and correctly chosen statistics methods for the validation of the result. However, the number of patients is fairly small, considering the incidence of the disease highlighted in the introduction and the number of years (5) taken into consideration, this hinders the value of the results. Also, the introduction may focus more on the impact of systemic inflammation upon vulvar cancer.

Author Response

Dear Reviewer 2,
The authors truly respect the time and effort you dedicated to providing constructive and insightful feedback on our manuscript. We appreciate the valuable input which allows us to revise our manuscript.
We outline our responses in answering the reviewer's comments and the detailed explanations in the attached document below. All the reviewer comments help us improve our manuscript and make it valuable to readers. 

Reviewer 3 Report

The study enrolls 86 patients with vulvar cancer over a period of 6 years. It would be advisable to correlate with other databases to increase the number of study patients. Also, these values obtained from the studied markers should be compared with vulvar cancer stages (FIGO, TNM, AJCC).

Taking into account that it is a retrospective study, there is a risk that the data collected may not be accurate with reference to other inflammatory factors that may be present in these patients. It would be advisable to conduct a prospective study so that these data are carefully selected.

It would also be indicated that the references should not be older than 10-15 years in order to be able to make a correct and up-to-date assessment of the discussions regarding this pathology.

Author Response

Dear Reviewer 3,
The authors truly respect the time and effort you dedicated to providing constructive and insightful feedback on our manuscript. We appreciate the valuable input which allows us to revise our manuscript.
We outline our responses in answering the reviewer's comments and the detailed explanations in the attached document below. All the reviewer comments help us improve our manuscript and make it valuable to readers. 
